# Telemedicine in Adult Congenital Heart Disease: Usefulness of Digital Health Technology in the Assistance of Critical Patients

**DOI:** 10.3390/ijerph20105775

**Published:** 2023-05-10

**Authors:** Nunzia Borrelli, Nicola Grimaldi, Giovanni Papaccioli, Flavia Fusco, Michela Palma, Berardo Sarubbi

**Affiliations:** Adult Congenital Heart Disease Unit, AO Dei Colli-Monaldi Hospital, 80131 Naples, Italy

**Keywords:** adult congenital heart disease, telemedicine, digital health

## Abstract

The number of adults with congenital heart disease (ACHD) has progressively increased in recent years to surpass that of children. This population growth has produced a new demand for health care. Moreover, the 2019 coronavirus pandemic has caused significant changes and has underlined the need for an overhaul of healthcare delivery. As a result, telemedicine has emerged as a new strategy to support a patient-based model of specialist care. In this review, we would like to highlight the background knowledge and offer an integrated care strategy for the longitudinal assistance of ACHD patients. In particular, the emphasis is on recognizing these patients as a special population with special requirements in order to deliver effective digital healthcare.

## 1. Introduction

As care for patients with congenital heart disease (CHD) has improved, the prevalence of this disease has shifted away from infancy and childhood toward adulthood. In 2008, the number of adults with CHD (ACHD) in the European Union exceeded for the first time that of children, and it is expected to continue to rise [1]. However, CHD is a lifelong condition, and appropriate follow-up in expert hands plays a key role in achieving a favorable long-term outcome by recognizing and addressing the specific and highly variable complications promptly [2].

Residual and progressive hemodynamic lesions, exercise intolerance, arrhythmias, and heart failure (HF) afflict many patients. As ACHD patients grow older, acquired cardiac and non-cardiac diseases and additional co-morbidities become increasingly present and relevant to the quality of life and late outcome. Adults with CHD continue to experience higher mortality rates than the general population, with patients with complex CHD, Fontan physiology, and Eisenmenger syndrome presenting the lowest survival rates [1].

The leading cause of death in complex ACHD is now HF. The management of this complication remains an ongoing challenge. HF can be subclinical, underscoring the need for tertiary follow-up. Timely catheter interventions and surgery have an established role in selected patients. Pharmacological treatment with continuously tailored dosages and eventual cardiac resynchronization therapy are recognized tools. Mechanical assist devices and heart transplantation remain options for end-stage patients when other management strategies have been exhausted.

Healthcare systems are able to accommodate and support the minority of patients requiring intensive care therapy. However, a vast group of ACHD patients require ongoing specialist assistance with daily monitoring of hemodynamic parameters to manage the typical instability of these conditions. Inpatient and outpatient clinic care cannot handle the large number of these patients. Tele- or video-clinics are now standard and have been well received by patients all over the world for a lot of medical specialties [3,4,5,6,7,8,9].

During the 2019 coronavirus (COVID-19) pandemic, telemedicine (TM) has shown the potentiality of substituting rather than integrating conventional face-to-face clinics. Many ACHD patients, who normally would attend tertiary centers for non-scheduled rapid assessment and care, might be managed at home under the close guidance and remote supervision of ACHD specialized teams, selecting those patients that must be admitted and receive tertiary inpatient care.

While digital health technology seems to be ideally suited for monitoring complex ACHD populations affected by HF and other comorbidities, there is very limited clinical experience with this technology in this population. As such a possibility needs to be clearly defined and organized, the purpose of this review is to summarize the most recent findings and suggest a patient-tailored strategy for the ongoing management of ACHD patients. Emphasis is placed on knowledge gaps and potential research topics to develop the best follow-up practices to support this unique population in living normal lives.

## 2. Adult Patients with Congenital Heart Disease: A Special Population with Special Needs

### 2.1. General Aspects of a Lifelong Chronic Condition

In recent years, dramatic improvements in medicine, surgery, and new technologies, ranging from fetal diagnosis to early complete repair and enhanced post-operative care, have resulted in a significant improvement in survival, allowing a greater proportion of patients with CHD to reach adulthood [10].

As survival and experience have progressed, it has become well-known that CHD is a lifetime condition with several complex healthcare needs. For most CHD patients, significant surgical sequelae and unusual hemodynamic features are expected. The main late complications and causes of hospitalization are cardiac arrhythmias, pulmonary vascular disease, infective endocarditis, thromboembolism, and HF, the latter also being the main cause of death [11]. Many patients may require further interventions, and figuring out the optimal timing for those procedures, ideally before any permanent organ damage, represents a major challenge for clinicians.

Apart from the cardiac problem itself, adults with CHD may present multiple comorbidities. Age-related exposure to risk factors is just as harmful to CHD patients as it is to non-CHD individuals. Primary prevention, with a thorough assessment and risk factor management approach, becomes essential in this population since the negative effects of overlapping acquired age-related conditions, such as diabetes, atherosclerosis, and arterial hypertension, may be amplified in this population. On the other hand, this population may suffer from chronic multi-organ deterioration, which may be related to the underlying CHD or specific repair and treatment and is commonly caused by long-term changes in hemodynamics, physiology, neurological, and psychosocial development. These complications can negatively affect patients’ quality of life and mortality and include liver and renal failure, lung disease, thyroid disorders, neurological problems, and erythropoietic system alterations [12,13,14].

Given the rising number of ACHD patients reaching childbearing age, more women with corrected, palliated, or uncorrected ACHD are experiencing pregnancy. Pregnancy may induce physiological changes, triggering HF, arrhythmias, and thrombotic complications in susceptible individuals. As a result, these changes may worsen maternal cardiac function, raising the risk of morbidity and mortality in both the mother and the fetus [15].

Finally, it should not be neglected that repeated hospital admissions and the need for continuous surveillance may have a huge negative impact on the quality of life of young ACHD patients.

### 2.2. Follow-Up in ACHD Patients

Considering the increasing proportion of adults with CHD, ensuring access to high-quality ACHD care is crucial. Specialized facilities, however, are not accessible everywhere. Patients often travel great distances to clinics, sustaining large costs for travel, accommodation, meals, and other related expenses. The standard of care at primary care facilities may be insufficient, with fewer health care resources and imaging facilities and a smaller variety of physicians and service providers [16].

In order to meet the specific needs of this population, it is essential to have a thorough understanding of the complex anatomy and deranged physiopathology of both repaired and unrepaired CHD. Hence, specialized follow-up by specifically trained personnel is paramount, especially for those with moderate to complex defects.

The 2020 European Society of Cardiology (ESC) guidelines recommend that all ACHD patients, including those with mild lesions, undergo at least one evaluation at an ACHD tertiary center and that an organized network of hub and spoke centers be established to care for ACHD patients [2]. To provide the highest level of assistance, tertiary ACHD centers must have clinicians specifically trained in the treatment of ACHD and advanced HF, personnel skilled in advanced imaging techniques, a service of congenital cardiac surgery, anaesthesiologists with experience in complex ACHD physiology, and interventional cardiologists able to perform invasive procedures on ACHD patients.

At the ACHD center, patients must be educated on their disease, emphasizing the role of regular expert follow-up. It is crucial to comprehend how having a cardiac problem affects daily life, especially for adolescents who are taking responsibility for their own healthcare and self-management. Education is paramount to empowering this young population. Patients must be trained to recognize “red flags” signs [such as fever, high heart rate (HR), and swelling legs] that should prompt medical contact. Programs of structured health education have been proven to raise patients’ awareness [17].

Some ACHD groups are more fragile and prone to sudden decompensation. A univentricular heart, either unrepaired or palliated with the Fontan procedure, represents one of the most challenging conditions [18]. In particular, in Fontan circulation, the systemic venous flow is redirected to the pulmonary branches. Thus, venous blood flow is entirely driven by the pressure gradient. These patients are particularly vulnerable to arrhythmic events and thrombotic complications, which may deteriorate their fragile balance and eventually culminate in refractory HF.

Moreover, patients with Fontan circulation may experience decreased oxygen saturation, particularly in the presence of fenestrated conduits or venovenous collaterals. Cyanosis is another significant feature involving numerous patients with complex ACHD or Eisenmenger syndrome. Chronic cyanosis is associated with a concomitant increased risk of bleeding complications and thrombosis, which is facilitated by secondary erythrocytosis, which results in increased blood viscosity and decreased flow velocity in the small arterioles and capillaries.

### 2.3. Lifelong Continuous Surveillance in ACHD: Lessons Learnt from COVID-19

The COVID-19 pandemic has transformed our lives and had a tremendous impact on healthcare delivery provisions. ACHD patients, in particular, have been shown to be susceptible to the potentially detrimental effects of the infection [19], especially those with complex physiology [20]. This unprecedented global crisis has challenged healthcare systems worldwide, forcing the urgent implementation of novel systems of care despite difficulties deemed insurmountable.

ACHD centers had to rapidly remodulate their models of care [21]. In this scenario, novel technologies applied to the healthcare system have evolved at a notably rapid pace. With the emergency of the pandemic under control, it is of utmost importance to continue to promote and invest in an integrated healthcare system, including remote monitoring of the most fragile ACHD patients.

## 3. Impact of Telemedicine on Health Care

Telemedicine is the electronic transmission of medical data from one location to another to provide patients with remote care. It can be delivered in different modalities. The synchronous method involves direct interaction between the interlocutors and the evaluation of the patient and clinical documentation in real time. Whereas the asynchronous method (“store and forward”) comprises a two-step interaction: firstly, data information and questions are collected, and secondly, answers and relative conclusions are delivered.

Several types of remote health services exist:Tele-examination: a medical procedure in which a doctor interacts remotely with a patient, eliciting a complete clinical history and performing a virtual physical examination;Tele-consultation: a remote consulting activity between a clinician and a patient that allows for the diagnosis or treatment of patients without their physical presence;Healthcare remote cooperation: assistance provided by a doctor or other healthcare professional to another engaged in a health act (e.g., guidance given during an emergency aid);Tele-monitoring: this technique involves the remote monitoring of clinical parameters using wearable, insertable, or close-proximity medical devices.

In recent years, because of the COVID-19 pandemic and the consequent reduction or interruption of several public health and hospital services, a variety of platforms have been created and specifically implemented to provide telehealth to patients via video teleconferencing [22,23].

Tele-examination and tele-consultation have been the best solutions during the pandemic for balancing patient safety, public health, and clinical efficacy in outpatient care. During tele-consultation, the patient can be linked to a virtual clinic and speak face-to-face with the cardiologist. The virtual visits may benefit from the visual clues offered by video images, the clinical data provided by patient-assisted maneuvers, and the implementation of various home monitoring devices [24].

Furthermore, considering the number of patients with implantable electronic devices (CIEDs) requiring follow-up exceeds the actual management capacity of hospitals, telemonitoring has become a necessity rather than a possibility.

Currently, the rapid development of wearable device technology along with CIEDs offers a practical option for:Device programming optimization; early detection of clinical problems or technical issues [25,26,27];Remote care of patients with HF, improving their clinical outcome and reducing the risk of recurrent hospitalizations and cardiovascular death [28];Mass screening and early diagnosis of both clinical and subclinical atrial fibrillation (AF) [29,30].

The “2021 ESC Guidelines on cardiac pacing and cardiac resynchronization therapy” recommend remote monitoring of CIEDs to reduce the number of in-office follow-ups, provide early detection of clinical problems or technical issues, and enable early recognition of actionable events in patients at greater risk (e.g., in the case of pacemaker dependency) [31].

Moreover, remote monitoring has shown many advantages over in-office follow-ups, including reduced costs for both hospitals and patients, reduced follow-up time, increased device longevity [32], reduced number of battery charges and inappropriate shocks resulting in fewer device replacements [33], reduced undetected arrhythmias, lead and generator malfunction [34], and stroke [35].

Among the CIEDs, the implantable cardioverter-defibrillators (ICDs), the cardiac resynchronization therapy defibrillators (CRT-Ds), and pacemakers (CRT-Ps) can detect several HF-related parameters predicting HF events (HFE) [36,37].

These parameters were used to create predicting HFE algorithms, such as:The heart logic is a multisensory algorithm combining first and third heart sounds, respiration rate, nocturnal HR, thoracic impedance, weight, and physical activity (sensibility 70%, specificity 85%, and a median lead time before HFE of 34 days) [37].The algorithm developed by D’Onofrio et al. includes temporal trends of diurnal and nocturnal HR, ventricular extrasystoles, atrial tachyarrhythmia burden, physical activity, and thoracic impedance (65.5% of first post-implant HF hospitalization prediction, a median alerting time of 42 days, and one false alert every 17 months) [38].Recently, multisensory non-invasive remote monitoring of physiological data by a chest-applied temporary patch has been shown to accurately predict hospitalization for HF exacerbation with 76–88% sensitivity, 85% specificity, and a median duration between initial alert and readmission of 6.5 days [39].

Another promising device for remote monitoring of HF patients is the CardioMEMS TM HF System. This device is implanted into the pulmonary artery and allows for real-time monitoring of pulmonary artery pressure, which represents an early warning indicator of worsening HF. Hemodynamic-guided HF management has been shown to reduce HF hospitalizations and improve patient-reported quality of life [40,41].

The “2021 ESC Guidelines for the Diagnosis and Treatment of Acute and Chronic Heart Failure” advocate wireless hemodynamic monitoring of pulmonary artery pressure in symptomatic patients with HF and left ventricular ejection fraction ≤ 35% [42].

Recently, an Israeli medical company, BioBeat Technologies Ltd. (Petah Tikva, Israel), has developed a monitoring system consisting of a wristwatch paired with a mobile application and a web platform. The wristwatch can continuously detect and transmit essential vital parameters, such as HR, HR variability, blood oxygen saturation, blood pressure, body temperature, sleep quality, cardiac index, movement sweat level, calories, cardiac output, stroke volume, systemic vascular resistance, and respiratory rate. Several studies [43,44,45,46,47,48,49,50,51] validated the reliability and validity of these measurements.

All these devices enable proactive and personalized patient management, allowing the prevention of HF exacerbation and subsequent hospitalization, with a net benefit for patients and the health system.

Telemonitoring is increasingly used in clinical practice, including for AF detection. Implantable cardiac monitors and mobile health (mHealth) wearable devices have been adopted for this purpose. While implantable cardiac monitors’ clinical value is well known, interest in the novel mHealth approach has been growing recently.

The mHealth technologies include photoplethysmography, single-lead ECG strips (iECG), and mechanocardiography.

The recently approved Food and Drug Administration-approved FibriCheck^®^ is a photoplethysmography-based application with a sensitivity of 83–96% and a specificity of 83–97% for AF detection [52].

Similarly, the iECG KardiaMobile (AliveCor^®^, Mountain View, CA, USA) has been approved by the Food and Drug Administration and presents a 71–99.6% sensitivity and 94.1–99.6% specificity for AF detection [53,54,55].

Many trials showed comparable results in sensitivity and specificity for AF detection using other wearable devices [56,57,58,59].

Overall, these devices have proven to be a valuable tool for the early diagnosis of AF, enabling timely and adequate prevention of thromboembolic events [33] and early therapy for rhythm control, preventing AF progression, and avoiding complications such as HF [60].

Moreover, this technology may improve the management of pre-existing AF by both following treatment response and defining the need for anticoagulation on the basis of AF burden and risk stratification [61].

Notably, Muhlestein et al. showed that the combination of serial smartphone single-lead recordings could create a virtual 12-lead ECG and identify ischemic changes such as ST elevation [62]. Detecting early signs of myocardial ischemia using telemonitoring tools could improve clinical outcomes, but further prospective clinical trials are needed [63].

## 4. Telemedicine in the Care of Adult Patients with CHD

While telemedicine is certainly an effective method for treating adult patients with CHD, there are still some challenges to be overcome. These include choosing the right patients to treat, selecting the parameters to monitor, and reducing the additional workload for healthcare staff.

A good patient selection process is probably the most crucial issue. Indeed, it would seem unrealistic and ineffective to simply apply TM to the entire population of CHD patients, as single ACHD centers often have to treat a large number of patients with various degrees of clinical status and severity of diagnosis [64]. In a certain way, the population of patients with complex CHD [2] appears to be the most suitable to be monitored by TM. However, additional factors need to be taken into consideration. Considering that the number of specialized centers for ACHD is often limited [65] and not sufficient for patients’ needs, many patients are forced to travel considerable distances, frequently making it impossible to adhere to the set follow-up schedule. In this regard, it would seem appropriate to extend the use of TM to patients who live far from the specialty facility and for whom it would become a crucial way of maintaining planned visits. Moreover, the majority of adults with CHD, even those with moderate or mild disease [9], have significant rates of healthcare utilization, especially for arrhythmias, which account for 37% of emergency admissions [66], and HF. Therefore, symptoms and frequent annual hospitalizations appear to be key elements to consider during patient selection.

Another crucial factor is choosing the parameters to monitor. The effectiveness of TM greatly depends on finding the most appropriate indicators of early deterioration. Several studies conducted on patients with HF have shown how the transmission of data such as body weight, blood pressure, oxygen saturation, and heart rhythm resulted in a lower risk of all-cause mortality [67,68]. It has also been demonstrated that remote rhythm monitoring of implantable cardiac defibrillators can reduce the occurrence of inappropriate shocks [69].

A further major aspect is the massive volume of patient-generated data. In order to prevent the loss of clinically significant data, it appears essential that particular healthcare professionals dedicate part of their working time to reviewing all the data, thus adding to the workload [64]. However, it is possible that information flow administration will be partially automated in the near future [70].

On the other hand, TM may provide many advantages in the care of ACHD patients by enabling a rapid out-of-hospital diagnosis, lowering mortality and morbidity, reducing general costs for the healthcare system, and increasing visit adherence.

Although adults with CHD may experience disorders due to general cardiovascular risk factors, such as coronary artery disease, the most frequent complications that result in morbidity and mortality are arrhythmias and HF [71].

Cardiac arrhythmias are a common symptom in ACHD patients and a significant contributor to morbidity and unplanned hospitalizations [66]. Since symptoms frequently do not manifest during Holter monitoring, the test can fail to yield a diagnosis. Telemedicine enables on-demand recordings of cardiac rhythm during palpitations, allowing for faster evaluation and treatment or remote reassuring [72]. Indeed, Ding et al. demonstrated that telemonitoring provides prompt out-of-hospital diagnosis and management of paroxysmal AF without the need for hospital admission [73].

Heart failure is responsible for approximately 20% of the deaths of ACHD patients, often during early adulthood [74]. In general, in patients with HF, remote health monitoring has been demonstrated to lower hospitalization and general mortality, while contrast results were demonstrated for the risk of all-cause hospitalization [67]. Moreover, for patients with acute myocardial infarction, in-hospital mortality was reduced when their electrocardiograms were remotely transmitted in advance [75].

Another appealing benefit of TM is the possibility of reducing costs. In a study by Ashwood et al. [76], Californian patients paid an average of USD 79 for telehealth consultations as opposed to USD 146 for an outpatient appointment. This is in addition to savings from travel-related costs and lost time, estimated at roughly USD 89 billion annually. Furthermore, in some cases, a home triage service can have the ability to divert patients from expensive emergency department care, saving about USD 1700 for each visit. Lastly, fewer hospitalizations can also lead to lower expenditures for the medical system.

Finally, telemonitoring seems well tolerated by adult patients with CHD. Indeed, ACHD patients are typically young people who enjoy using mobile technology and can easily adhere to a video consultation program without significant challenges. Adult patients with CHD have demonstrated a good general adherence rate to TM programs, with higher program adherence among patients who present clinical events during follow-up [72]. In general, TM has demonstrated an overall improvement in patients’ attendance at medical appointments, both in person and remotely [77].

## 5. Discussion

The burden of CHD has been changing worldwide, and the proportion of adults with CHD has now reached and surpassed that of children throughout the industrialized regions of the world [78]. As a result of the decreased mortality rate and increased survival of patients with CHD, a growing need for access to healthcare and continued monitoring, especially for severe forms of CHD, has emerged.

Moreover, the spread of the COVID-19 pandemic in 2020 has emphasized the need for ongoing care for fragile patients with lifelong comorbidities [20] and urged improving healthcare digitalization with the development of remote monitoring tools, as well as mitigating home-to-hospital travel and virus exposure [79]. As a result, in the last few years, ACHD centers have experienced an exponential increase in remote monitoring capabilities, ranging from telephone calls and multimodality virtual visits to pacemaker-defibrillator monitoring and minimally invasive long-term implantable rhythm monitoring.

TM has improved the ability of cardiologists to offer high-quality care to an ever-growing number of adult patients with CHD. Greater effectiveness and non-location-specific expert assessments have contributed to this advancement.

Modern technological developments have had a significant impact and facilitated the growth of TM to its current position. The adoption of wireless and wearable technology, in particular, has enhanced patient monitoring, allowing for rapid action in the event of any parameter deviation from normal values.

As already demonstrated in HF [67,68], telemonitoring variables such as body weight, blood pressure, oxygen saturation, and HR may impact the prognosis even for critically ill ACHD patients. Current advancements in TM devices have enabled further evaluation of variables such as cardiac index, cardiac output, stroke volume, and systemic vascular resistance. In complex ACHD patients who chronically suffer from low blood oxygen saturation and low blood pressure, these particular assessments may anticipate the patient’s decompensation and promote preemptive measures.

Moreover, the use of advanced technologies, attention to patient history, connectivity with dedicated databases, and prediction systems may impact outcomes, improving the quality of management and ultimately helping to save lives.

Telemedicine has been shown to be highly effective and well-tolerated by patients with ACHD. It supports physicians in providing patient-centered care by respecting and responding to patients’ unique preferences, requirements, and values and by ensuring that patient needs guide all clinical choices [80]. It enables early diagnosis of arrhythmias and arterial hypertension and detects early signs of patient decompensation, helping to minimize delays in the delivery of assistance and treatment to frail patients with ACHD and resulting in an overall decrease in hospitalizations and deaths [72]. Moreover, adherence to visits has improved, in particular when distance or time represent obstacles or make ACHD clinics inaccessible to patients [81]. All in all, considering the worldwide spread of digital technologies, mobile devices, and personal computers, TM allows for enhanced care delivery and an inclusive level of assistance, ensuring that the standard of treatment is not affected by a patient’s socioeconomic status, gender, or geographical location [79].

## 6. Future Directions

Considering healthcare assistance for ACHD patients is organized nowadays using the hub and spoke model, which considers the existence of frequently significant geographic distances between the patient’s residence and the tertiary centers, TM can enable a balance between health promotion, optimization of hospital resources, and patient compliance [82]. Indeed, TM may facilitate the alternation between remote and in-person assistance based on the preferences and needs of doctors and patients.

Frequent virtual appointments, between occasional “in the Hub” follow-up, may be planned for patients at moderate to high risk of complications, who have a low awareness of heart disease, and/or who live far from the ACHD centers. Further assessments may be arranged at the spoke centers using teleconsultation.

In this context, we believe artificial intelligence may offer effective resources for monitoring patients’ vital signs, allowing early identification of possible health issues, assisting in diagnosis, providing personalized therapies, sending appointment reminders, and improving general adherence to drugs.

Such integrated assistance would enable the transition to continuous rather than occasional care and drive toward a patient-tailored care model (Figure 1).

## 7. Conclusions

Telemedicine can represent a new model of care for ACHD patients at high risk of repeated hospital admissions. Essential elements of the model should be: personalized ACHD care with patient and family education; fast access to ACHD databases with automatic notifications for specific actions; daily closer surveillance of hemodynamic parameters; continuous phone and video relationship; the possibility of fast access to attend the ACHD hub directly from home because of signs of instability; and timely catheter interventions and surgery (central illustration).

This assistance model can finally take into account the quality of life of these complex ACHD patients, who can have tailored, highly-specialized care directly from home. We must do better on this front.

## Figures and Tables

**Figure 1 ijerph-20-05775-f001:**
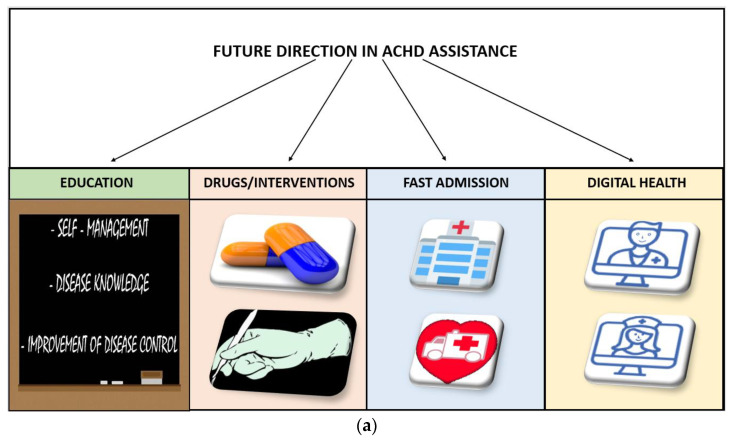
(**a**) Future direction in Adult Congenital Heart Disease assistance. (**b**) Central illustration.

## Data Availability

No new data were created or analyzed in this study. Data sharing is not applicable to this article.

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
