# Peer review of "Telemedicine in Adult Congenital Heart Disease: Usefulness of Digital Health Technology in the Assistance of Critical Patients"

_ijerph, 2023, doi:10.3390/ijerph20105775_

Round 1
Reviewer 1 Report
Effective telemedicine requires choosing appropriate indicators of deterioration in heart failure patients, such as body weight, blood pressure, oxygen saturation, and heart rhythm, as demonstrated by several studies. In discussion section, it’s important to discuss that (1) such indicators need to monitor plus (2) if similar nature of indicators of other participant’s data may available for better and quick diagnoses which is need to telemedicine too.
Telemedicine can provide personalized and closer surveillance of high-risk ACHD patients' hemodynamic parameters, improving their quality of life but (3) in-depth usage of user-facing technologies and patient history maintenance and connectivity with related databases and predictions system may produce improved results. Need to consider this in the discussion section.
Overall it’s a well-explanatory article.
Reviewer 2 Report
A brief summary
This study proposes the use of telemedicine teams for the care of patients with congenital heart disease by focusing on adult patients with congenital heart disease. The paper's ideas and recommendations are of more useful value for the use of telemedicine in response to sudden illness.
Comment 1
Line 47-48
The authors attempt to explore the use of telemedicine in the treatment of patients with congenital heart disease. In the absence of current studies on telemedicine for patients with congenital heart disease, I suggest that the authors provide other examples of telemedicine for similar conditions as a reference.
Comment 2
Line 98-99
The authors present the general characteristics and medical risks of patients with congenital heart disease. However, the authors do not elaborate on the impact of reasons such as accessibility and the effectiveness of health education on patients with congenital heart disease. I suggest that the authors add the impact of management issues in the health care system on patients with congenital heart disease.
Comment 3
3.3. Impact of telemedicine on health care
The authors mention the impact of telemedicine services on patients with congenital heart disease. However, the narrative only focuses on the role of telemonitoring, but not on the role of teleexaminations and consultations. I suggest that the authors could add a description of the role of telemedicine and consultation.
Round 2
Reviewer 2 Report
I am satisfied with the authors' response.